# KINet: Keypoint Interaction Networks for Unsupervised Forward Modeling

## Abstract

Object-centric representation is an essential abstraction for physical reasoning and forward prediction. Most existing approaches learn this representation through extensive supervision (e.g., object class and bounding box) although such ground-truth information is not readily accessible in reality. To address this, we introduce KINet (Keypoint Interaction Network)—an end-to-end unsupervised framework to reason about object interactions in complex systems based on a keypoint representation. Using visual observations, our model learns to associate objects with keypoint coordinates and discovers a graph representation of the system as a set of keypoint embeddings and their relations. It then learns an action-conditioned forward model using contrastive estimation to predict future keypoint states. By learning to perform physical reasoning in the keypoint space, our model automatically generalizes to scenarios with a different number of objects, and novel object geometries. Experiments demonstrate the effectiveness of our model to accurately perform forward prediction and learn plannable object-centric representations which can also be used in downstream model-based control tasks.

## 1 Introduction

Discovering a structured causal representation of the world allows humans to perform a wide repertoire of motor tasks such as interacting with objects. At the core of this process lies the ability to predict the response of the environment to applying an action (Miall & Wolpert, 1996; Wolpert & Kawato, 1998). Such an internal model, often referred to as the forward model, aims to come up with an estimation of future states of the world given its current state and an action. By cascading the predictions of a forward model it is also possible to plan a sequence of actions that would bring the world from an initial state to a desired goal state (Wolpert et al., 1998; 1995).

Recently, various deep learning architectures have been proposed to perform forward modeling using an object-centric representation of the system (Ye et al., 2020; Chen et al., 2021b; Li et al., 2020; Qi et al., 2020). This object-centric representation is learned from the visual observation by factorizing the scene into the underlying object instances using ground-truth object states (e.g., object class, position, and bounding box).

We identified two major limitations in the recent work: First, existing approaches either assume access to the ground-truth object states (Battaglia et al., 2016; Li et al., 2020) or predict them using idealized techniques such as pre-trained object detection or instance segmentation models (Ye et al., 2020; Qi et al., 2020). However, obtaining ground truth objects state information is not feasible in practice. Relying on object detection and segmentation tools, on the other hand, makes the forward model fragile and dependent on the flawless performance of these tools. More often than not, pre-trained object detection or segmentation models suffer from poor generalization to unseen objects. Second, factorizing the scene into object instances limits the generalization of forward models to scenarios with a different number of objects.

In this paper, we address both of these limitations by proposing to learn forward models using a keypoint-based object-centric representation. Keypoints represent a set of salient locations of moving entities. Our model KINet (Keypoint Interaction Network) learns unsupervised forward modeling in three major steps: (1) A keypoint extractor factorizes the scene into keypoint coordinates with no supervision other than raw visual observations. (2) A probabilistic graph representation of the system is inferred where each node corresponds to a keypoint and edges are keypoints relations.

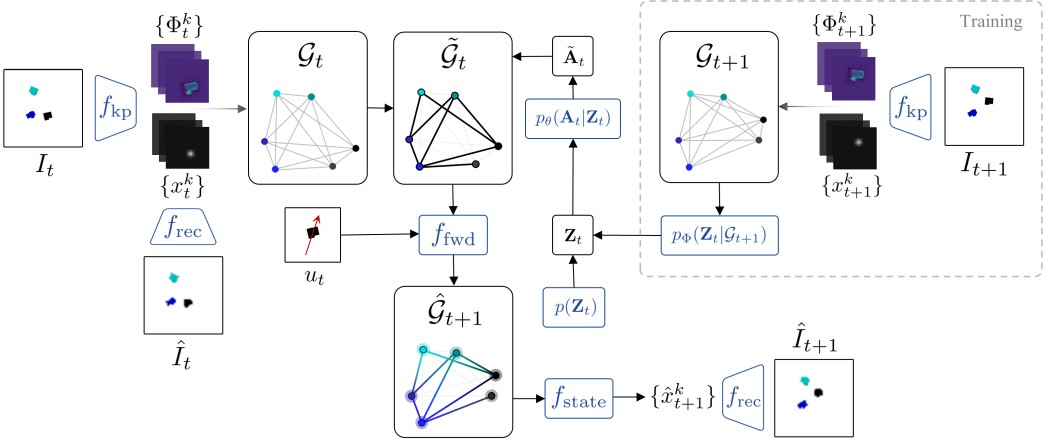

Figure 1: KINet Architecture. Our model performs forward modeling based on image observations in three major steps: extracting keypoint coordinates, inferring a probabilistic graph representation of the system, and estimating the next state of the system conditioned on the action. Learned functions and distributions are the blue blocks.

Therefore, each object is represented in this graph by at least one distinct node. Node features carry implicit object-centric representation as well as explicit keypoint state information. (3) With probabilistic message passing, our model performs physical reasoning and learns an action-conditional forward model to predict future locations of the keypoints. Using this prediction, it also reconstructs the future appearances of the system. We evaluate KINet's forward prediction accuracy and demonstrate that, by learning the physical reasoning in a keypoint coordinate, our model can effectively re-purpose this knowledge and generalize it to complex unseen circumstances.

Our key contributions are as follows: (1) We introduce KINet, an end-to-end method for learning unsupervised action-conditional forward models from visual observations using keypoint coordinates. (2) We introduce Probabilistic Interaction Networks for efficient message-passing in fully connected graphs by learning to adaptively aggregate a subset of relevant neighborhood information. (3) We introduce the GraphMPC algorithm for accurate action planning based on graph similarity. (4) We demonstrate that learning forward models in keypoint coordinates enables zero-shot generalization to complex unseen scenarios.

## 2   RELATED WORK

**Unsupervised keypoint extraction** methods have been used successfully in computer vision tasks such as pose tracking (Zhang et al., 2018; Yao et al., 2019) and video prediction (Minderer et al., 2019; Zhang et al., 2018; Xue et al., 2016). Recent work explored keypoint extraction for control tasks in reinforcement learning environments to project the visual observation space to a lower-dimensional keypoint space (Kulkarni et al., 2019; Chen et al., 2021a; Jakab et al., 2018). We extend these methods for the object manipulation setting.

**Forward prediction** models estimate the future state of a system given its current state. There is a large line of work on using neural networks to build such models for various applications. These approaches date back to Nguyen & Widrow (1990); Grzeszczuk et al. (1998) for modeling state transition in dynamic systems. The most fundamentally relevant approach to our model is the Interaction Network (IN) (Battaglia et al., 2016; Sanchez-Gonzalez et al., 2018) and other follow-up work on using graph neural network in forward modeling (Pfaff et al., 2020; Li et al., 2019; Kipf et al., 2018; Mrowca et al., 2018). Although these approaches demonstrated powerful forward modeling capabilities, they often rely on building explicit object representations based on ground-truth state information of the objects in the system. In an attempt to bridge this gap several methods have been proposed to use IN-based models using visual features extracted from image observations. (Watters et al., 2017; Qi et al., 2020; Ye et al., 2020). However, two main concerns remain unaddressed in these methods. First, the visual features of objects are often extracted from object bounding boxes

using pretrained object detection or segmentation model (Janner et al., 2018; Qi et al., 2020; Kipf et al., 2018) that is either pretrained on the environment (Qi et al., 2020) or assume prior knowledge of the object position (Ye et al., 2020). Second, implementing these approaches entails training the model on a fixed number of objects.

**Learning action-conditional forward models** is an active challenge although the original IN incorporates the action as an external effect by aggregating the action vector to node and edge embeddings. For the case of probabilistic forward models Henaff et al. (2019) interestingly suggests using a latent variable dropout to ensure proper conditioning of the forward model on the action information following Gal & Ghahramani (2016). In a more relevant application to our model, Yan et al. (2020) highlighted the effectiveness of contrastive estimation (Oord et al., 2018) to learn proper plannable object-centric representations.

**Graph representation inference** methods has been originally introduced in variational graph autoencoders by Kipf & Welling (2016). To formulate the graph structure inference module we were also inspired by Simonovsky & Komodakis (2018) where they use a variational graph autoencoder to generate small graphs of molecular structures. More relevant to our work is Kipf et al. (2019) in which an object-level contrastive loss has been used to learn object-centric abstractions in multibody systems. However, this work is only restricted to fixed environments with oversimplified objects such as two-dimensional geometries (Watters et al., 2019; Stanić et al., 2020). In our work, we consider experimenting with more complex three-dimensional objects where the objects are randomized and replicate real-world object manipulation.

## 3 KEYPOINT INTERACTION NETWORKS (KINET)

We assume access to observational data that consists of images, actions, and resulting images after applying actions: $\mathcal{D} = \{(I_t, u_t, I_{t+1})\}$ where $I_t, I_{t+1} \in \mathbb{R}^{H \times W \times C}$ are images before and after applying action and $u_t \in \mathbb{R}^4$ is the action vector. Our goal is to learn a forward model that predicts future states of the system with no supervision above the observational data. In this section, we describe our approach which is composed of two main steps: learning to encode visual observations into keypoint coordinates and then learning an action-conditioned forward model in the keypoint space to reason about the system and predict its future state.

### 3.1 UNSUPERVISED KEYPOINT DETECTION

The keypoint detector $f_{\mathrm{kp}}$ is a mapping from visual observations to a lower-dimensional set of $K$ keypoint coordinates $\{x_t^k\}_{k=1\ldots K} = f_{\mathrm{kp}}(I_t)$. The keypoint coordinates are learned through capturing the spatial appearance of the objects in the system in an unsupervised manner. The detector design is following the idea of Kulkarni et al. (2019); Jakab et al. (2018), which we extend for object manipulation setting that necessitates consideration of external actions.

Specifically, The keypoint detector receives a pair of initial and current image frames $(I_0, I_t)$ and uses a convolutional encoder to compute a $K$-dimensional feature map for each image $\Phi(I_0), \Phi(I_t) \in \mathbb{R}^{H' \times W' \times K}$. The expected number of keypoints in the system is captured by the dimension $K$. Next, each keypoint feature map is marginalized into a 2D keypoint coordinate $\{x_0^k, x_t^k\}_{k=1\ldots K} \in \mathbb{R}^2$. We use a convolutional image reconstruction model $f_{\mathrm{rec}}$ with skip connections to inpaint the current image frame using the initial image and the predicted keypoint coordinates $\hat{I}_t = f_{\mathrm{rec}}(I_0, \{x_0^k, x_t^k\})$.

With this formulation, $f_{\mathrm{kp}}$ and $f_{\mathrm{rec}}$ create a bottleneck to encode the visual observation in a temporally consistent lower-dimensional keypoint coordinate representation which is distributed across the visual observation.

### 3.2 GRAPH REPRESENTATION OF SYSTEM

The primary idea of our approach is representing the system as a graph. In particular, after factorizing the system into $K$ keypoints, we build a directed graph $\mathcal{G}_t = (\mathcal{V}_t, \mathcal{E}_t, \mathbf{A}_t)$ where keypoints become the graph nodes and their pairwise relations become the graph edges. Keypoint positional and visual information are encoded into the feature embeddings of nodes $\{\mathbf{n}_t^k\}_{k=1\ldots K} \in \mathcal{V}_t$ and

edges $\{\mathbf{e}_t^{ij}\} \in \mathcal{E}_t$. We also use an adjacency matrix to further specify the graph connectivity as $\mathbf{A}_t \in \mathbb{R}^{|\mathcal{V}| \times |\mathcal{V}|}$ where $\{\mathbf{e}^{ij}\} \in \mathcal{E}$ if $[\mathbf{A}_t]_{ij} = 1$.

At each timestep $t$ the observational data is encoded into the graph. Node embeddings are an encoding of the keypoint visual features and positional information extracted in the keypoint detector $\{\mathbf{n}_t^k\} = [x_t^k, \Phi_t^k]$. Edge embeddings contain a relative positional information of the sender and receiver nodes $\{\mathbf{e}_t^{ij}\} = [x_t^i - x_t^j, \|x_t^i - x_t^j\|_2^2]$ as suggested by Pfaff et al. (2020). We assume that there is no self-loop in the graph nodes (i.e, diagonal elements in the adjacency matrix is set to 0). Note that the graph representation is built on keypoints coordinates and, hence, we do not impose any prior assumption on the number of objects in the system.

## 3.3 PROBABILISTIC INTERACTION NETWORKS

To build a forward model we extend the recent approaches based on Graph Neural Networks (Battaglia et al., 2016; Sanchez-Gonzalez et al., 2018; Pfaff et al., 2020) and propose a probabilistic variation of the Interaction Networks (IN). The core of the probabilistic IN is generating node-level latent variables $\mathbf{Z}_t \in \mathbb{R}^{|\mathcal{V}| \times d}$ (Kipf & Welling, 2016; Simonovsky & Komodakis, 2018). The node-level prior latent variables are independently sampled from a fixed isotropic Gaussian prior distribution $p(\mathbf{Z}_t)$.

During training, the posterior distribution of the latent variable is obtained from the next timestep graph,

$$q_\phi(\mathbf{Z}_t | \{\mathbf{n}_{t+1}^k\}_{k=1...K}) = f_{\text{enc}}(\mathcal{G}_{t+1}) \tag{1}$$

We use a probablistic decoder to derive the posterior probability of the adjacency matrix given the latent variable. To build this decoder, the existance of edges in the graph is modeled as Bernoulli variables with probabilities defined as,

$$p_\theta([\mathbf{A}_t]_{ij} = 1 | \mathbf{z}_t^i, \mathbf{z}_t^j) = \sigma(f_{\text{dec}}(\mathcal{G}_t, \mathbf{Z}_t)) \tag{2}$$

where $\sigma(\cdot)$ is the sigmoid function. We propose to use the posterior adjacency matrix distribution along with the latent variables to infer a probabilistic graph representation of the system $\tilde{\mathcal{G}}_t = (\tilde{\mathcal{V}}_t, \tilde{\mathcal{E}}_t, \tilde{\mathbf{A}}_t)$. In particular, the inferred adjacency matrix $\tilde{\mathbf{A}}_t \sim p_\theta(\mathbf{A}_t | \mathbf{Z}_t)$ includes the edge probabilities. The node embeddings are also appropriately aggregated with the node-level latent variables $\{\tilde{\mathbf{n}}_t^k : [\mathbf{n}_t^k, \mathbf{z}_t^k]\}_{k=1...K} \in \tilde{\mathcal{V}}_t$.

The probabilistic IN forward model $\hat{\mathcal{G}}_{t+1} = f_{\text{fwd}}(\tilde{\mathcal{G}}_t, u_t; \mathbf{Z}_t, \tilde{\mathbf{A}}_t)$ predicts the graph representation at the next timestep by taking as input the current probabilistic graph and action. The message-passing operation in the forward model can be described as,

$$\{\hat{\mathbf{e}}^{ij}\} \leftarrow f_e(\tilde{\mathbf{n}}^i, \tilde{\mathbf{n}}^j, \tilde{\mathbf{e}}^{ij}), \quad \{\hat{\mathbf{n}}^k\} \leftarrow f_n(\tilde{\mathbf{n}}^k, \sum_{i \in N(k)} \hat{\mathbf{e}}^{ik}, u_t) \tag{3}$$

where the edge-specific function $f_e$ first updates edge embeddings, then the node-specific function $f_n$ updates node embeddings using neighboring edge information. Note that the message-passing operation takes into account the node-level latent variables and the neighborhood aggregation $N(k)$ is also performed based on the probabilistic adjacency matrix.

Recent models for forward prediction often rely on fully connected graphs for message passing (Qi et al., 2020; Ye et al., 2020; Li et al., 2018). Our model, however, learns to dynamically sample the neighborhood of each node at each timestep conditioned on the latent variable. Intuitively, this adaptive sampling allows the network to efficiently aggregate long-range context information only by selecting the most relevant neighboring nodes. This is specifically essential in our model since keypoints could provide redundant information if they are in very close proximity.

## 3.4 FORWARD PREDICTION

Using the predicted graph representation $\hat{\mathcal{G}}_{t+1}$, we interpret a first-order difference of the keypoint states. The state decoder $f_{\text{state}}$ transforms the predicted node embeddings to a first-order difference which is integrated once to predict the position of the keypoints in the next timestep $\{\hat{x}_{t+1}^k\} = \{x_t^k\} + f_{\text{state}}(\{\hat{\mathbf{n}}_{t+1}^k\})$.

To reconstruct the image at the next timestep we borrow the reconstruction model $f_{\text{rec}}$ from the keypoint detector. The initial image along with predicted keypoint states at the next time step produces an estimated rendering of the future appearance of the system $\hat{I}_{t+1} = f_{\text{rec}}(I_0, \{x_0^k, \hat{x}_{t+1}^k\}_{k=1...K})$.

### 3.5 LEARNING KINET

**Reconstruction loss.** The keypoint detector is trained using the $\mathcal{L}_2$ distance between the ground truth image and the reconstructed image at each timestep $\mathcal{L}_{\text{rec}} = \|\hat{I}_t - I_t\|_2^2$. As suggested by Minderer et al. (2019), errors from the keypoint detector were not backpropagated to other modules of the model. This is a necessary step to ensure the model does not conflate errors from image modules and reasoning modules.

**Inference loss.** Our model is also trained to infer the adjacency matrix. This goal is acheived by optimizing the evidence lower bound (ELBO) (Kipf & Welling, 2016). This includes maximizing the likelihood of the adjacency matrix given the latent distribution and minimizing the KL-divergence between the posterior and prior latent distribution:

$$\mathcal{L}_{\text{infer}} = \mathbb{E}_{q_\phi(\mathbf{Z}|\mathcal{G})}[-\log p_\theta(\mathbf{A}|\mathbf{Z})] + D_{\text{KL}}\big(q_\phi(\mathbf{Z}|\mathcal{G}) \, \| \, p(\mathbf{Z})\big) \tag{4}$$

We take independent Gaussian prior $p(\mathbf{Z}) = \prod_i \mathcal{N}(\mathbf{z}_i)$ and posterior $q_\phi(\mathbf{Z}|\mathcal{G}) = \prod_i \mathcal{N}(\mathbf{z}_i|f_{\text{enc}}(\mathcal{G}))$ distributions for node latent variables and use reparameterization trick for training (Kingma & Welling, 2013).

**Forward loss.** The model is also optimized to predict the next state of the keypoints. A forward loss penalizes the $\mathcal{L}_2$ distance between the estimated future keypoint locations using first-order state decoder and the keypoint extractor's prediction:

$$\mathcal{L}_{\text{fwd}} = \sum_K \|\hat{x}_{t+1}^k - f_{\text{kp}}(I_{t+1})^k\|_2^2 \tag{5}$$

**Contrastive loss.** In our model, we are seeking to learn representations to build an action conditioned forward model. Therefore, to further encourage the model to learn actionable object-centric representations, we make use of the contrastive estimation method. We add a contrastive loss as described in Oord et al. (2018); Yan et al. (2020) and reframe it for graph embeddings as:

$$\mathcal{L}_{\text{ctr}} = -\mathbb{E}_\mathcal{D}[\log(\frac{\mathcal{S}(\hat{\mathcal{G}}_{t+1}, \mathcal{G}_{t+1}^+)}{\sum \mathcal{S}(\hat{\mathcal{G}}_{t+1}, \mathcal{G}_{t+1}^-)})] \tag{6}$$

where $\mathcal{S}$ is a graph matching algorithm. With this loss, we maximize a lower bound on the mutual information of the learned graph representations. Specifically, we want to ensure that the predicted graph representations $\hat{\mathcal{G}}_{t+1}$ are maximally similar to their corresponding positive sample pairs $\mathcal{G}_{t+1}^+$ and maximally distant from the negative sample pairs $\mathcal{G}_{t+1}^-$. We use a simple node embedding similarity as the graph matching algorithm $\mathcal{S}(\mathcal{G}_1, \mathcal{G}_2) = \sum_K \{n_1^k\}.\{n_2^k\}$.

Finally, the combined loss of the model can be written as:

$$\mathcal{L} = \lambda_{\text{rec}} \, \mathcal{L}_{\text{rec}} + \lambda_{\text{fwd}} \, \mathcal{L}_{\text{fwd}} + \lambda_{\text{infer}} \, \mathcal{L}_{\text{infer}} + \lambda_{\text{ctr}} \, \mathcal{L}_{\text{ctr}} \tag{7}$$

### 3.6 GRAPHMPC PLANNING WITH KINET

We use a learned KINet model and plan actions based on a Model Predictive Control (MPC) algorithm (Finn & Levine, 2017) in the graph embedding space (GraphMPC). We sample several actions at each timestep to apply them to the current graph representation of the system. With KINet forward prediction, we compute a predicted graph representation for the next timestep given a sampled action. We then take the optimal action that produces the most similar graph representation to a goal graph representation $\mathcal{G}^{\text{goal}}$. We describe our GraphMPC algorithm with a time horizon of $T$ as:

$$u_t^* = \arg\max_{u_t}\{\mathcal{S}\big(\mathcal{G}^{\text{goal}}, f_{\text{fwd}}(\mathcal{G}^t, \{u_{t:T}\})\big)\}; \; t \in [0, T] \tag{8}$$

where $\mathcal{S}$ is a graph matching algorithm. Unlike performing conventional MPC only with respect to positional states, GraphMPC allows for accurately bringing the system to a goal state both explicitly (i.e, position) and implicitly (i.e, pose, orientation, and visual appearance).

# 4 EXPERIMENTAL SETUPS

Our experiments are motivated by the following questions: (1) Does the model accurately learn a forward model? (2) Can we use the model to implement action planning? and (3) How well our model generalizes to unseen circumstances?

## 4.1 ENVIRONMENT

We apply our approach to learn a forward model for multi-object manipulation tasks. The task involves rearranging multiple objects in the scene and bringing them to a desired goal state using point-to-point pushing actions. We use MuJoCo 2.0 (Todorov et al., 2012) to simulate training and testing scenarios.

Specifically, we generate a total of 10K episodes of random object manipulations where multiple objects (1-5 objects) are present in the scene and a simple robot end-effector applies randomized pushing actions for 90 timesteps per episode. In the simulation, each object is represented as a combination of 2 cuboid geoms with randomized length, width, and color to diversify the objects. We uniformly sample the randomized length and width from a predefined continuous range ($\text{geom}_{train}$ for training and $\text{geom}_{gen}$ for generalization to unseen geometries, see Appendix A for more details). At each timestep, we only collect the 4-dimensional action vector (pushing start and end location) and RGB images of the scene before and after the action is applied. Images are obtained using an overhead camera (*Top View*) and an angled camera (*Angled View*) (see Fig 3).

## 4.2 BASELINES

We compare our approach with previous methods on learning object-centric forward models:

**Forward-Inverse Model** *(ForwInv)*: We train a convolutional encoder to extract visual features of the scene image (*Img*) and jointly learn forward and inverse models in the feature space following Agrawal et al. (2016).

**Interaction Network** *(IN)*: We follow Battaglia et al. (2016); Sanchez-Gonzalez et al. (2018) to build an Interaction Network based on the ground truth location of the objects. Each object is represented with a vector representation that contains the ground-truth position and velocity of the objects (*GT state*). This approach is only applicable to scenarios where the number of objects in the scene is known and fixed.

**Visual Interaction Network** *(VisIN)*: We train a convolutional encoder to extract visual features of fixed-size bounding boxes centered on ground-truth object locations (*GT state + Img*). We use the extracted visual features as object representations in the Interaction Network. This approach also requires prior knowledge of the number of objects in the scene (Kipf et al., 2019; Watters et al., 2017).

## 4.3 TRAINING AND EVALUATION SETTING.

All models are trained on a subset of the simulated data where 3 objects are present in the scene which gives a total of approximately 8K episodes. We split this dataset into training (80%), validation (10%), and testing (10%) sets. We report the performance of our model using the testing data. To evaluate generalization to a different number of objects, we use other subsets of data with 1, 2, 4, and 5 objects (approximately 400 episodes for each case). We also measure generalization for unseen geometries where we sample object length and width out of the training dimension range. We train our model separately on images obtained from the overhead camera (*Top View*) and the angled camera (*Angled View*).

Table 1: Forward Prediction performance measured in position error for a single-step prediction. The prediction errors are computed for models separately trained on *Top* and *Angled View* images. Our model accurately predicts the next timestep using only image observations.

| | | Mean Position Error $\times 10^{-3}$ | |
|---|---|---|---|
| Model | Supervision | *Top View* | *Angled View* |
| ForwInv | Img | $0.293_{\pm 0.08}$ | $0.266_{\pm 0.02}$ |
| IN | GT state | $0.112_{\pm 0.003}$ | $0.109_{\pm 0.008}$ |
| VisIN | GT state + Img | $0.107_{\pm 0.006}$ | $0.121_{\pm 0.03}$ |
| KINet (Ours) | Img | $0.122_{\pm 0.01}$ | $0.129_{\pm 0.02}$ |
| KINet - deterministic | Img | $0.127_{\pm 0.03}$ | $0.133_{\pm 0.01}$ |
| KINet - no ctr loss | Img | $0.173_{\pm 0.02}$ | $0.169_{\pm 0.05}$ |

## 5 RESULTS

This section is organized to answer a series of questions to thoroughly evaluate our model and justify the choices we made in formulating our approach.

### 5.1 DOES THE MODEL ACCURATELY LEARN A FORWARD MODEL?

First, we evaluate if our model can accurately perform a single-step forward prediction. The prediction error is computed as the average $\mathcal{L}_2$ distance between the predicted and ground truth positional states. Note that in the baseline models, the number of objects in the scene is known and fixed ($N = 3$). However, our model does not make any assumption on the number of objects. Instead, we only set the expected number of keypoints in the system ($K = 6$).

We demonstrate the effectiveness of our model in comparison with ForwInv, IN, and VisIN baseline performances (Table 1). We separately train and examine the performance of each model on *Top View* and *Angled View* images. Among baseline models, VisIN performs best since it uses ground-truth object position information along with an encoding of the object visual features to build object representations. Our model, on the other hand, achieves a comparable performance to VisIN while it does not rely on any supervision beyond the scene images.

We also compare our model against ForwInv baseline that has similar supervision for training and show that our model produces forward predictions that are significantly more accurate. This emphasizes the capability of our approach to accurately learn a forward model while relaxing prevailing assumptions of the prior methods on the structure of the system and availability of ground-truth state information.

### 5.2 CAN WE USE THE MODEL IN CONTROL TASKS?

To further compare our approach with baseline methods, we use our model to predict a T-step sequence of pushing actions to bring objects to a desired goal state using an MPC planning based on *Top View* observations. For all models, we perform 1000 trials where at each trial object geometries and initial pose are randomized and a random goal configuration of objects is also generated. The planning horizon is set to $T = 40$ timesteps. For our model, we perform GraphMPC based on graph embedding similarity as described in Section 3.6. For all baseline models, we perform MPC directly based on the distance to the goal.

Figure 2 shows MPC results over the planning horizon. Our approach is consistently faster than baseline models in reaching the goal configuration. Additionally, we compute the pixel-wise distance to the goal image to evaluate whether the planned actions reach the goal configuration in terms of object appearances (i.e, orientation).

### 5.3 DOES THE MODEL GENERALIZE TO UNSEEN CIRCUMSTANCES?

One of our main motivations to learn a forward model in keypoint coordinate space is to eliminate the dependency of model formulation to the number of objects in the system. We test for zero-

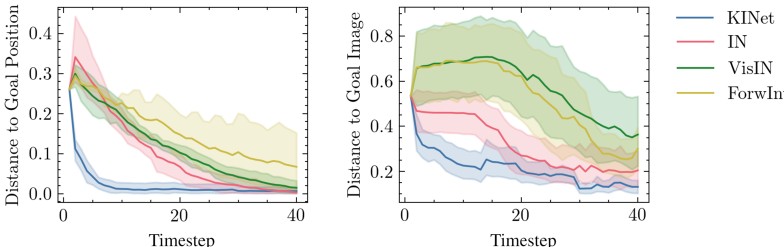

Figure 2: MPC comparison with baseline models for *Top View* images. Distance to goal measured based on of (a) position and (b) pixel errors. Comparison with baseline models shows that our model is faster and more accurate for planning.

shot generalization to a different number of objects (1, 2, 4, and 5 objects) and unseen geometries ($\text{geom}_{gen}$). See Figure 3 for qualitative generalization results. We separately train and examine for generalization on *Top View* and *Angled View* observations. Note that baseline models are not able to generalize to a different number of objects because this number is hard-coded in their formulation (e.g., in the number of feature channels or the number of graph nodes).

For generalizations, we set the planning horizon to $T = 80$. Since our model learns to perform forward modeling in the keypoint space, with zero-shot generalization, it reassigns the expected keypoints ($K = 6$) to unseen objects and then makes forward prediction. We observed that the keypoint extraction is, however, slightly less consistent compared to the trained scenario with 3 objects.

We also quantitatively evaluate the quality of generalization scenarios (Table 2) and as expected by increasing the number of object the average distance to goal position increases. Also, objects with geometries that were sampled out of the training dimension range has more distance to the goal position.

## 5.4 ANALYSIS AND ABLATION

We further justify the major choices we made to formulate the model by conducting ablation studies. Specifically, we examine two elements in our approach: the probabilistic graph representation, and the contrastive loss. We train two variants of KINet: (1) *KINet - deterministic* in which the graph representation is not probabilistic and there's no inference on the graph adjacency matrix. (2) *KINet - no ctr* where we train the model without adding the contrastive loss.

The best forward prediction performance for both *Top View* and *Angled View* observations is achieved when the model is probabilistic and trains with a contrastive loss (Table 1). The contrastive loss is an essential element in our approach to ensure the learned forward model is accurately action conditional. Also, a probabilistic graph representation allows for more efficient information aggregation of neighboring nodes. With a probablistic graph representation our model achieves higher generalization performance compare to its deterministic variant. This performance gap is more evident when generalizing to unseen geometries (Table 2).

## 6 CONCLUSION

In this paper, we proposed a general method for learning action-conditioned forward models based on image observations of the system. We showed that our approach effectively performs physical reasoning by inferring the structure of the system. Additionally, we demonstrated that a keypoint-based forward model makes fewer assumptions about the system. This in turn allows for an automatic generalization to a variety of unseen circumstances. Our method addresses a frequent issue with the prior work by learning an accurate forward model without explicit supervision on ground-truth object information. An interesting future direction is to focus on the keypoint extraction sim2real gap to further help with building reliable forward models for real settings. Finally, we

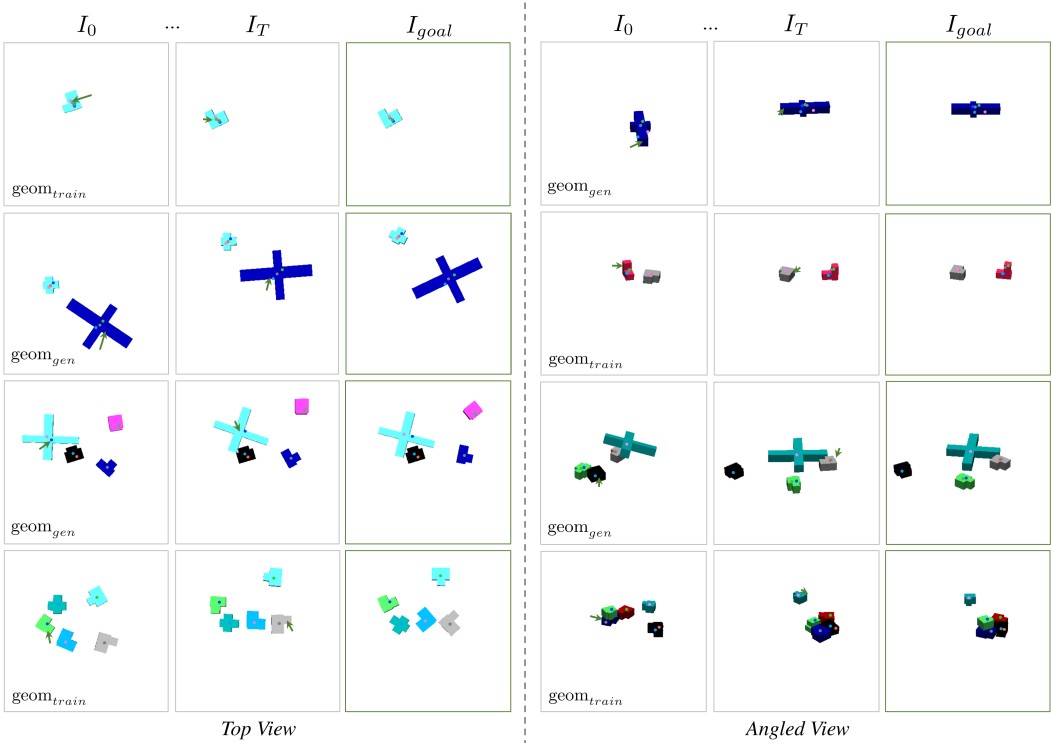

Figure 3: Qualitative results of generalization to different number of objects and unseen geometries for *Top* and *Angled View* observations. With zero-shot generalization, our model assigns keypoint to the objects and performs forward modeling in unseen scenarios. The green arrows depict the optimal action found with GraphMPC at each timestep. See Appendix C for detailed results.

Table 2: Generalization results measured by the average distance to the goal position for different number of objects and unseen object geometries in *Top* and *Angled View* images. Using a probabilistic graph representation significantly improves the generalization.

| | | \multicolumn{2}{c}{Distance to Goal Position $\times 10^{-3}$} | | | |
| | | \multicolumn{2}{c}{KINet} | \multicolumn{2}{c}{KINet - deterministic} | |
| | Object | $\text{geom}_{train}$ | $\text{geom}_{gen}$ | $\text{geom}_{train}$ | $\text{geom}_{gen}$ |
|---|---|---|---|---|---|
| Top View | 1 | $0.24_{\pm 0.02}$ | $0.31_{\pm 0.01}$ | $0.25_{\pm 0.02}$ | $0.34_{\pm 0.05}$ |
| | 2 | $0.22_{\pm 0.01}$ | $0.58_{\pm 0.02}$ | $0.26_{\pm 0.02}$ | $0.65_{\pm 0.03}$ |
| | 3 | $0.18_{\pm 0.03}$ | $0.21_{\pm 0.01}$ | $0.19_{\pm 0.06}$ | $0.28_{\pm 0.08}$ |
| | 4 | $0.54_{\pm 0.01}$ | $0.63_{\pm 0.13}$ | $0.68_{\pm 0.09}$ | $0.89_{\pm 0.11}$ |
| | 5 | $0.86_{\pm 0.08}$ | $1.73_{\pm 0.16}$ | $0.94_{\pm 0.06}$ | $2.01_{\pm 0.14}$ |
| Angled View | 1 | $0.21_{\pm 0.04}$ | $0.35_{\pm 0.04}$ | $0.28_{\pm 0.01}$ | $0.36_{\pm 0.08}$ |
| | 2 | $0.21_{\pm 0.03}$ | $0.53_{\pm 0.06}$ | $0.22_{\pm 0.05}$ | $0.59_{\pm 0.07}$ |
| | 3 | $0.19_{\pm 0.02}$ | $0.20_{\pm 0.05}$ | $0.19_{\pm 0.03}$ | $0.31_{\pm 0.08}$ |
| | 4 | $0.51_{\pm 0.02}$ | $0.65_{\pm 0.07}$ | $0.57_{\pm 0.12}$ | $0.96_{\pm 0.09}$ |
| | 5 | $0.89_{\pm 0.13}$ | $1.64_{\pm 0.11}$ | $1.05_{\pm 0.16}$ | $2.17_{\pm 0.10}$ |

hope our general approach inspires future research on physical reasoning in settings where ground-truth information is hard to obtain.

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

## A    DATA AND MUJUCO ENVIRONMENT

To generate simulated dataset we represent each object with two cuboid-shaped geoms. We initialize each simulation episode by randomize the geom size, color, and pose. We define the training geom dimensions as $\text{geom}_{train}$ with a width range in $[0.02, 0.04]$ and a length range in $[0.02, 0.06]$. We also generate a dataset with elongated geoms for generalization experiments with $\text{geom}_{gen}$ length in range $[0.06, 0.18]$. All geom heights are fixed to $0.03$. We simulate random 1-step point-to-point pushing actions in range $[0.01, 0.05]$ with random initialize position in proximity of the objects $[0.01, 0.03]$ towards uniformly sampled directions.

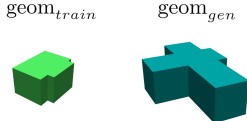

Figure 4: Example of objects for training and generalization. Each object is a combination of two cuboid geoms with randomly sampled dimensions.

## B    LEARNED GRAPH REPRESENTATION

Here, we include examples the learned graph representation with our model. Figure 5 shows an example of the inferred probabilistic graph adjacency matrix. Although it is not trivial how the structure of the scene is reflected in the graph connectivity, our generalization results shows that a probabilistic graph representation enables the model to better generalize to unseen geometries (see Table 2).

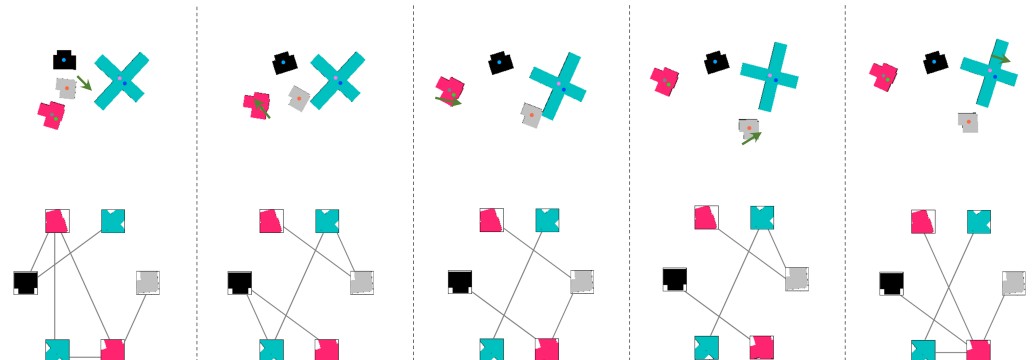

Figure 5: Detected keypoints and the inferred graph connectivity. Note that here we are plotting the edges with probability of more than 0.5; however, the graph representation of the scene is a probabilistic fully-connected graph.

To demonstrate the effectiveness of our approach for learning meaningful object-centric representation of the system we compute 2D t-SNE embeddings of the learned node features in our model. In these examples we can see that the learned features of the nodes that belong to the same object form distinct clusters (Fig 6).

## C    DETAILED MPC RESULTS

Here, we include a more detailed qualitative results of the MPC planning steps using our model for *Top View* (Fig 7) and *Angled View* (Fig 8) observations. Using our model in the downstream control task results in faster and more accurate planning.

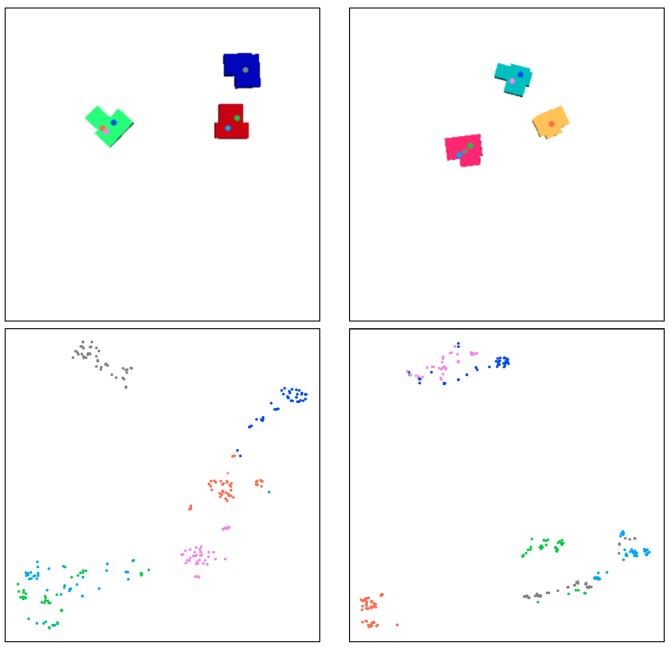

Figure 6: Node features 2D t-SNE plots.

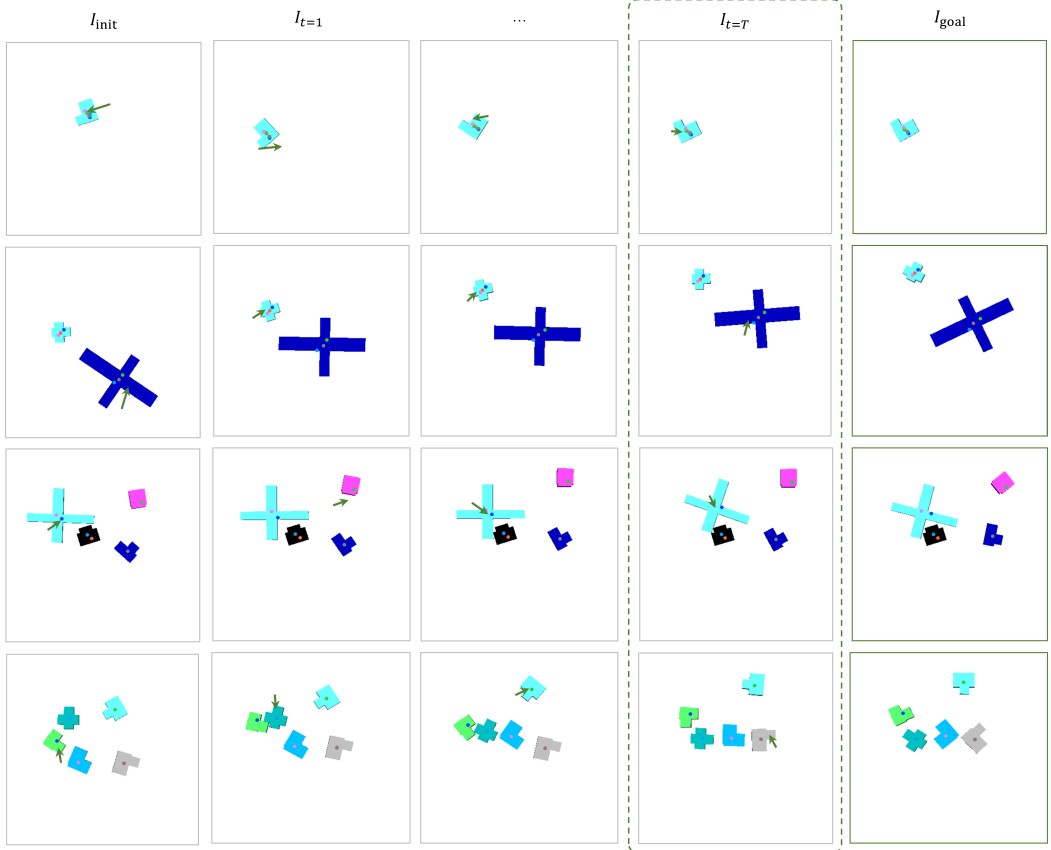

Figure 7: MPC results steps for *Top View* observations.

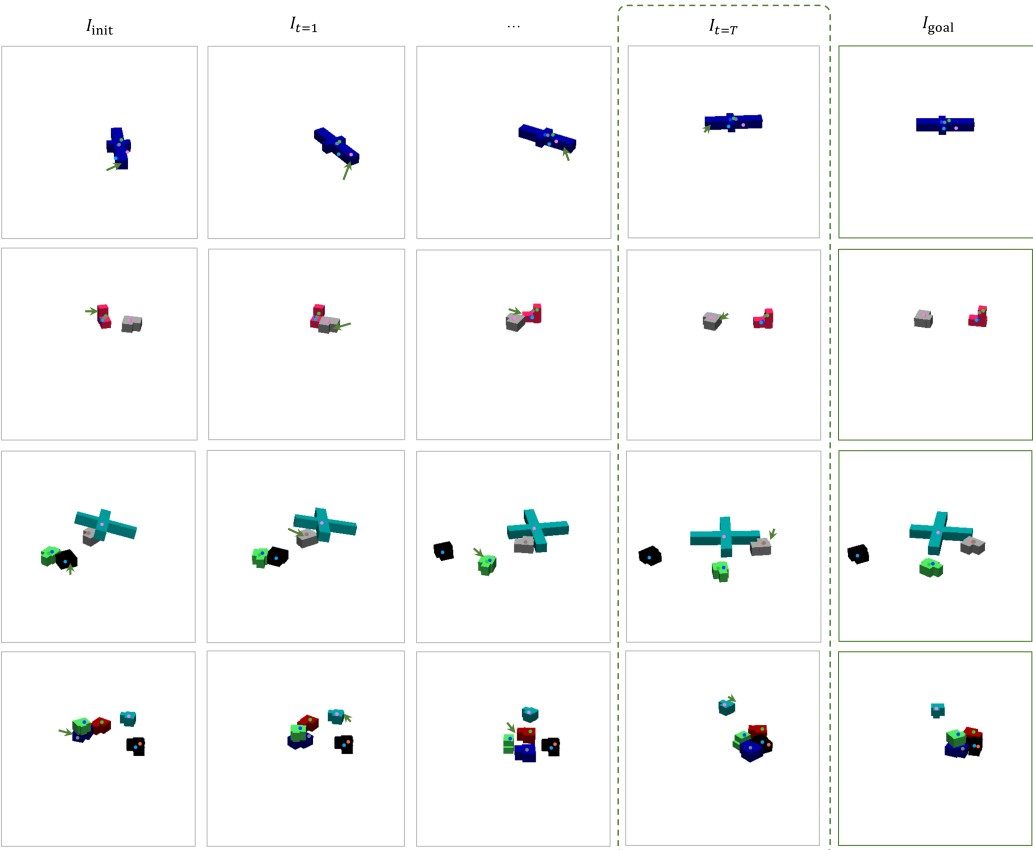

Figure 8: MPC results steps for *Angled View* observations.

