# OpenReview forum: "KINet: Keypoint Interaction Networks for Unsupervised Forward Modeling"
_ICLR.cc/2022/Conference — ICLR 2022 Submitted_

### Official Review · Reviewer_58w4 · 2021-10-22

**Correctness:** 3
**Technical Novelty And Significance:** 2
**Empirical Novelty And Significance:** 2
**Recommendation:** 3
**Confidence:** 4

**Main Review:**

Strength:
This work propose to achieve multi-object position prediction with only using visual information like images without ground truth positions as supervision.

Weakness:
1 In general, this work is a patchwork of a condition-based generative model and a raw image preprocessing method at the input and output ends, leading to insufficient novelty.
   1) For key point detection, Jakab et al. (2018) thoughts are followed.
   2) The initial graph Gt with its nodes and edges is built through widely used operations, e.g., visual+position; distance
   3) Why the approximate posterior is conditioned on the graph at next step? There is no interpretation, but this design is seem as some condition-VAE-based multi-agent prediction methods [1,2], while the only difference is that existing works infer the agents' dynamics but KINet infers the graph representation.
   4) The message passing is not novel.
   5) The forward prediction with a skip connection is not novel.

2 Why do you use the contrastive loss? The interpretation is not clear; why it helps to learn actionable object-centric representation? How to build the negative G-?

3 As for the experiments, I do not think the dataset matches the authors' claim: hard to annotation/detection and complex scenes.
   1) The objects and scenes are quite simple. I suggest to test the methods on some real-world dataset like SDD and ETH-UCY, which have top-view image and scene information like obstacles.
   2) In the proposed dataset, I notice that there are very large `+' object, but the method only determines a few key points for each object. So how to simulate the positional constraints imposed by the shape of an object？



[1] It Is Not the Journey But the Destination: Endpoint Conditioned Trajectory Prediction (ECCV2020)
[2] Contextually Plausible and Diverse 3D Human Motion Prediction (ICCV 2021)

**Summary Of The Paper:**

This paper introduces KINet (Keypoint Interaction Network)---an end-to-end unsupervised framework to reason about object interactions in complex systems based on a keypoint representation. There are three major steps in this model: extracting keypoint coordinates, inferring a probabilistic graph representation of the system, and estimating the next state of the system conditioned on the action.

**Summary Of The Review:**

This work propose to achieve multi-object position prediction with only using visual information like images without ground truth positions as supervision. But the detailed designs are not sufficiently novel.

---

> ### Author Response · Authors · 2021-11-25
> **Response to Reviewer 58w4**
>
> We thank the reviewer for their time and feedback. Several changes to the manuscript the been made including a new dataset and further experiments. We believe that through your assistance the paper has improved.
>
> >  Why do you use the contrastive loss? How to build the negative G-?
>
> Prior work showed the effectiveness of a contrastive learning framework for learning action-conditional forward models [1]. We also observed a piece of similar evidence in our ablation results (see Table 1). Without the contrastive loss, the accuracy of our action-conditioned forward model drops significantly.
>
> To obtain negative samples, within each simulation episode we set $G_{t+1}^{-}:= G_{\tau} \ \ \forall \tau \neq t+1$. In other words, in each simulated episode, we are applying a random sequence of pushing action so negative samples for each timestep are selected from other timesteps in the same episode (with similar randomized objects).
>
> > As for the experiments, I do not think the dataset matches the authors' claim.
>
> This is a valid concern and we agree that overhead camera observations were simplistic and essentially 2D. We added a side-view camera to our MuJoCo environment and generated a new 3D dataset. We repeated all experiments in the paper for both top-view and angled-view. The Results section is updated accordingly.
>
> > Weakness: 1 In general, this work is a patchwork of a condition-based generative model and a raw image preprocessing method at the input and output ends, leading to insufficient novelty.
>
> Thank you for bringing up this point. We do agree that the basis of our method is combining and repurposing preexisting ideas. Our approach brings together ideas from computer vision (unsupervised keypoint extraction) and machine learning (graph-based forward models). As also pointed out in other reviews, the idea of combining unsupervised keypoint detection and forward modeling is novel and has not been explored before. We also believe that our novel formulation of a probabilistic interaction network improves the existing graph-based forward models. Our results show powerful generalization to an unseen number of objects and unseen geometries.
>
>
> [1] "Learning Predictive Representations for Deformable Objects Using Contrastive Estimation" Wilson Yan, et al, 2020.

---

> > ### Comment · Reviewer_58w4 · 2021-11-26
> > **There is not any answer to the most crucial question 1 about novelty.**
> >
> > There is not any answer to the most crucial question 1 about novelty.
> >
> > As for the contrastive loss, some experience in the existing works or experimental results would not indicate the rationality in the proposed design. In other words, I still consider that, you just expand the optimization directions but without sufficient interpretation of 'actionable object-centric representation'.

---

> > > ### Author Response · Authors · 2021-11-26
> > > **Response is Updated**
> > >
> > > We sincerely apologize for this mistake. We had intended to provide a response to your first concern. The response is now updated.
> > >
> > > For our claims on the contrastive loss, we agree that we need to tone down our conclusion for interpreting that this loss has specifically helped conditioning the learned function on the action. Our results support the claim that the action-conditioned forward model is more accurate through a contrastive learning framework which is in line with findings in prior work (Wilson Yan, et al, 2020). We also used our forward model in a control task and performed model predictive control by sampling action inputs to our model and showed accurate control tasks results (see Figure 2 and Table 3). We conclude that our model would have not been able to achieve accurate performance in the control tasks had it not been properly conditioned on the action.

---

### Official Review · Reviewer_nqXG · 2021-10-31

**Correctness:** 2
**Technical Novelty And Significance:** 3
**Empirical Novelty And Significance:** 3
**Recommendation:** 5
**Confidence:** 4

**Main Review:**

Strengths:
- The paper is well-organized
- The problem motivation in the introduction is clearly explained and analysis of related work is quite thorough
- I appreciate that the experiment section includes not only experiments on reconstruction and future prediction, but also a simple control task. This is important to show the applicability of the proposed method.

I am generally positive about this paper. However, the main weakness is the characterization of the method in section 3. While reading, I could not find the justification for some claims and/or could not match the textual description with the notation in the formulas. Here is a list of major points in order of appearance:
- Section 3.2 (and following): how is the adjacency matrix defined? If no external information about the environment is used, then $\[A_t\]_ij=1\ \forall t,i,j$, i.e. the graph is assumed fully-connected. If this is the case, why focussing on reconstructing the graph structure (likelihood maximization in eq5). All adjacency matrices, both initial and reconstructed, will be full of ones anyway, right? It should be noted that this was not an issue in Kipf&Welling 2016, which the paper indicates as the base for KINet, since the VGAEs were applied to graphs with a non-trivial adjacency matrix.
- KINet remains rather inflexible wrt the number of keypoints, even though the paper tries to present it as more flexible and generalizable than other baselines and related works:
  - Section 3.2: "Note that the graph representation is built on keypoints coordinates and, hence, we do not impose any prior assumption on the number of objects in the system." but actually the number of keypoints is fixed by the number of feature maps K in $f_{kp}$.
  - Section 5.3: "baseline models are not able to generalize to a different number of objects because this number is hard-coded in their formulation (i.e, in the number of feature channels or the number of graph nodes)". Actually, KINet uses a constant K in this exact way, the only difference is that K is set to 6 which is greater than the max number of objects used in the experiments.
- The method is strongly related to Kipf&Welling 2016, as mentioned often in the paper, but at the same time there are strong inconsistencies. For example, why is the node-level prior (eq 1) conditioned on $G_t$? In Kipf&Welling the prior is a zero-centered unit-variance normal distribution that does not depend on the graph. The same definition of prior, including the dubious dependency on the graph, is repeated in section 3.5.
- Also related to the point above, eq2 uses the graph at the next timestep $G_{t+1}$ to obtain $Z_t$. Since the training is based on predicting the next graph based on the current, how is this choice justified? Why doesn't it interfere with training? Also, if eq2 is used during training as mentioned in the text above the equation, what is the behavior during inference when the future is not accessible?
- The neighborhood aggregation mechanism that appears in eq4 is portrayed as a main contribution of the paper (see introduction). However, I could find neither a definition nor an ablation study of such a component. The only vague explanation is at the end of section 3.3: "Our model, however, learns to dynamically sample the neighborhood of each node at each timestep conditioned on the latent variable". To be considered a main contribution, I would expect a more detailed description of what $N(k)$ means and how it affects the model.
- Since the method focuses strongly on defining a graph representation for the keypoints it's weird that in the experiment section this graph structure is not inspected at all. Considering that a key component of the method is inferring the adjacency matrix $\tilde{A}_t$, it would be good to analyze the learned matrices, characterize them in terms of sparsity, and verify whether they capture semantic information about the environment.
- How are negative samples $G^-$ for the contrastive loss (eq7) defined? On the same topic, how is the claim "The contrastive loss is an essential element in our approach to ensure the learned forward model is action conditional" in 5.4 justified? How does the contrastive loss relate to the model being action conditional?
- Also on the contrastive loss, can S be considered a graph matching algorithm? First, the algorithm assumes the same number and order of nodes between the two graphs. Isn't it possible that the encoder network outputs two isomorphic graphs that are identical expect in the order of the nodes? It would be enough for the feature maps $\phi_k$ to specialize in specific regions of the input image. Also, the graph matching algorithm S completely disregards the adjacency matrices. The two graphs could have similar node embeddings but very different connectivity and they would be considered similar. This further shows how the graph representation and the probabilistic inference of the adjacency matrix might not be a key component of the method.

Other weaknesses:
- Using L2 distance as a reconstruction loss in image space will not scale well to higher-dimensional domains such as natural images. Of course, this paper doesn't have to solve every problem with representation learning, but since this limits the applicability of the method it should at least be mentioned.

Other questions:
- Why does the keypoint detector take as input the pair $(I_0, I_t)$ as opposed to $(I_{t-1}, I_t)$? What is the advantage in using the initial state of the system? Since the simulated tasks start from random configurations with no inherent initial state, how does the choice of $I_0$ affect the detection of keypoints, the reconstruction, and the forward model? How far in the future can the forward model run?
- Is the dataset of 10K episodes described in 4.1 available for reproducibility?
- Is the test set of 1000 control tasks used in 5.2 released?
- What is the size (e.g. side length) of the simulation used in the experiments? This is important to understand if the mean position errors reported in the tables are small or large in absolute terms.

Minor points:
- Figure 1: repeated word "are are"
- Section 2: "relate work" -> "related work"
- Figure 2: x axis labels "timetep" -> "timestep", also improve font size
- Section 5.2: is "converging" the right word here? Convergence sounds like a model being trained, but my understanding is that the MPC algorithm uses the learned forward model to predict the outcome of future actions and take one step. I suggest replacing with "reaching" that gives the idea of achieving the desired state following a trajectory.
- Section 5.3: "i.e. in the number of feature channels or the number of graph nodes" should be "e.g."
- Figure 3: the reader might benefit from a little more details in caption. What does each row represent? What are the dots and arrows in the images?

**Summary Of The Paper:**

The authors introduce Keypoint Interaction Network, an unsupervised framework for keypoint-based representation learning and forward modelling.
This framework is accompanied by auxiliary algorithms for training and quantitative experiments for evaluation.

**Summary Of The Review:**

The paper gives much importance to a graph-based representation of interacting systems. However, the proposed formulation doesn't seem to leverage the graph in full. Actually, it might even be possible to describe the entire method without discussing nodes and adjacency matrices.

Some elements of the method are not convincing, e.g. prior function, graph matching, reconstruction loss. Other components, e.g. the contrastive loss, are under-specified. Some claims are not well supported: flexibility of KINet wrt the parameter K and effect of the contrastive loss.

Overall, the method of the paper sounds promising, but I don't recommend it for acceptance in its present form. I suggest polishing the method, re-evaluating the role of the graph representation, revising some vague claims, clarifying some definitions and formulas, and incorporating more experiments and ablation studies.

---

> ### Author Response · Authors · 2021-11-25
> **Response to Reviewer nqXG  [1/3]**
>
> We thank the reviewer for their time and their many insightful feedback and constructive suggestions. Several changes to the manuscript the been made including a new dataset and further experiments. We believe that through your assistance the paper has improved.
>
> ### Method Characterization
> >  ... how is the adjacency matrix defined?
>
> The adjacency matrix is defined as a fully connected graph. We proposed to make the decoder (see equation 2) output a probabilistic fully connected graph. The decoder takes as input the node embeddings and their latent variables and outputs the probability over the edges in the graph (as in [1,2]).
>
> > All adjacency matrices, both initial and reconstructed, will be full of ones anyway, right?
>
> This is an important point. Our probabilistic graph decoder function infers the likelihood of the adjacency matrix by $\sigma (f_{dec}(G_t, Z_t))$. This is different from [1] since our decoder is also taking into account the node embeddings and is not just conditioned on the latent variable. We think this is potentially the reason that the inferred graphs in our results are more sparse and not full of ones. We included an example in Appendix B.
>
> > KINet remains rather inflexible wrt the number of keypoints ...  actually the number of keypoints is fixed by the number of feature maps K in $f_{kp}$ ... the only difference is that K is set to 6 which is greater than the max number of objects used in the experiments.
>
> This is a subtle observation. We agree that setting fixed $K$ keypoints can be seen as another hardcoding issue in formulating the problem. However, with our approach, we show that a fixed number of keypoint feature maps $K$ is in fact more flexible than fixing the number of objects in the formulation as in prior work. This flexibility enables the model to achieve zero-shot generalization to unseen geometries and an unseen number of objects (see Figure 3).
>
> > The method is strongly related to [1] ... why is the node-level prior (eq 1) conditioned on $G_t$?
>
> Thank you for bringing this to our attention. We apologize for the confusing notations we used in this section that caused this ambiguity. As implied in Figure 1, we are also using fixed isotropic Gaussian prior distributions. To clarify this, we revised the poor choice of notation in our first draft and rewrote some of the equations in Sections 3.3 and 3.5.
>
> > ... $p_{\theta}$ uses the graph at the next timestep $G_{t+1}$ to obtain $Z$
>
> During training the latent variables $Z$ are sampled from the posterior network $q_{\phi}$ conditioned on the next timestep graph $G_{t+1}$. This network is trained based on the total loss by minimizing the KL distance between the posterior distribution and the fixed prior. For inference, latent variables are sampled from the prior distribution.
>
> > The (probabilistic) neighborhood aggregation mechanism ... I could find neither a definition nor an ablation study of such a component.
>
> We apologize for this confusion. The neighborhood aggregation in our approach is performed by summation of neighboring node embeddings in the probabilistic fully connected graph. The neighbor information aggregation is a weighted sum of the neighbor nodes based on the inferred edge probabilities. We performed a series of ablation studies to compare our proposed Probabilistic Interaction Network with a deterministic neighborhood aggregation mechanism (which is commonly used in graph-based forward models [6]). We found that our probabilistic framework achieves the highest generalization performance to an unseen number of objects and unseen object geometries (see Figure 3 and Table 2).
>
>
> > It would be good to analyze the learned adjacency matrices, characterize them in terms of sparsity, and verify whether they capture semantic information about the environment.
>
> We agree that this is a good addition to the results. We added an example of the inferred graphs in the Appendix section. In Figure 5, although the learned graph representations are probabilistic and fully connected, we obtained discrete estimation of graph connectivity by filtering out the edges with a probability of less than 0.5. In Figure 6, we demonstrate 2D t-SNE embedding of the learned graph representations.
>
> Although the inferred connectivities in the graphs do not trivially reflect the structure of the scene, we found strong semantic information about the environment in the node features. We observed that the learned node features that belong to the same object form distinct clusters which implies that the learned graph representation captures a strong object-centric representation of the environment. Also, we added additional ablation experiments to compare the probabilistic graph representation with deterministic fully connected graph representation and found the probabilistic framework significantly improves generalization to an unseen number of objects and unseen geometries (see Table 2).

---

> ### Author Response · Authors · 2021-11-25
> **Response to Reviewer nqXG (cont)  [2/3]**
>
> > How are negative samples $G^{-}$ for the contrastive loss (eq7) defined? ... How does the contrastive loss relate to the model being action conditional?
>
> Prior work showed the effectiveness of a contrastive learning framework for learning action-conditional forward models [3]. We also observed a piece of similar evidence in our ablation results (see Table 1). Without the contrastive loss, the accuracy of our action-conditioned forward model drops significantly.
>
> To obtain negative samples, within each simulation episode we set $G_{t+1}^{-}:= G_{\tau} \ \ \forall \tau \neq t+1$. In other words, in each simulated episode, we are applying a random sequence of pushing action so negative samples for each timestep is selected from other timesteps in the same episode (with similar randomized objects).
>
> Importantly, this choice for defining negative samples requires careful implementation otherwise it potentially results in an infinite loss (equation 6). In the simulated data, actions are randomly sampled around the objects but for those timesteps where the action does not change the configuration of the scene (for instance when the randomly sampled pushing action does not engage with any of the objects in the scene) the similarity loss defined in equation 6 could fall into a division by a very small ($\exists t: G_{t+1}^{-} \simeq G_{t+1}$). To prevent this, we defined a keypoint location distance threshold for selecting the negative samples to ensure that the negative samples are selected from the timesteps where action changed the configuration of the scene.
>
> >  Isn't it possible that the encoder network outputs two isomorphic graphs that are identical expect in the order of the nodes?
>
> This is a great point. In the keypoint extraction architecture, we used $f_{kp}$ learns to discover temporally consistent keypoints [4,5]. The order of graph nodes corresponds to the order of the extracted keypoints which remain consistent across timesteps due to this temporal consistency.
>
> > The graph matching algorithm S completely disregards the adjacency matrices ... probabilistic inference of the adjacency matrix might not be a key component of the method.
>
> We agree that we could have used alternative graph matching algorithms to incorporate the adjacency matrix in building the contrastive loss and also the GraphMPC algorithm. To avoid overcomplicating our formulation we started from the simpler case of only computing the graph matching based on node embeddings. Our model, with the existing graph matching algorithm, learned an accurate forward model with successful performance in downstream MPC control tasks.
>
> Moreover, we believe the advantage of learning probabilistic graph representation is more evident in generalization experiments. We expanded our ablation experiments and found that probabilistic graph representation significantly enhances zero-shot generalizability to an unseen number of objects and object geometries (see the updated Figure 3 and Table 2).
>
> > Using L2 distance as a reconstruction loss in image space will not scale well to higher-dimensional domains such as natural images.
>
> Computing the reconstruction loss with L2 norm for unsupervised keypoint extraction also appears in almost all prior work [3,4,7]. We are not aware of other specific alternatives that would improve the reconstruction.

---

> ### Author Response · Authors · 2021-11-25
> **Response to Reviewer nqXG (cont) [3/3]**
>
> ### Other questions
> > Why does the keypoint detector take as input the pair ($I_0, I_t$) as opposed to $(I_{t-1}, I_{t})$?
>
> This is an interesting question. We have tried both and decided to choose ($I_0, I_t$) because with this pair $f_{kp}$ achieved higher reconstruction performance and qualitatively it also found more temporally consistent keypoints. We think $I_{t}$ and $I_{t+1}$ are likely to be very similar if the randomly sampled pushing action $a_{t}$ does not change the configuration of the objects significantly. Using ($I_0, I_t$), on the other hand, provides two distinct appearances of the scene and is a better source of information for unsupervised keypoint extraction.
>
> > How far in the future can the forward model run?
>
> This is a good suggestion. For this paper, we have not focused on future rollouts but we would like to consider it for our follow-up work. According to Table 1, the 1-step forward prediction is accurate and we expect our model to also have acceptable future rollouts accuracy.
>
> > Is the dataset of 10K episodes described in 4.1 available for reproducibility? Is the test set of 1000 control tasks used in 5.2 released?
>
> Our dataset will be released upon the acceptance of the paper which could serve as a useful resource for future work with a focus on the action-conditional forward modeling. Following the recent free release of the latest version of MuJoCo, we are also planning on releasing the MuJoCo simulation environments we used to generate the dataset.
>
> > What is the size (e.g. side length) of the simulation used in the experiments?
>
> We added Appendix A to elaborate on the details of our MuJoCo simulation environment. This section gives an overview of the object geometries and dimensions in the training and generalization datasets.
>
> > Minor points.
>
> We really appreciate these suggestions. We have now fixed these typos.
>
> [1] "Variational Graph Auto-Encoders." Thomas N Kipf, Max Welling, NeurIPS 2016.
> [2] "GraphVAE: Towards Generation of Small Graphs Using Variational Autoencoders", Martin Simonovsky, Nikos Komodakis, 2018.
> [3] "Learning Predictive Representations for Deformable Objects Using Contrastive Estimation" Wilson Yan, et al, 2020.
> [4] "Causal Discovery in Physical Systems from Videos." Li et al., NeurIPS 2020.
> [5] "Unsupervised Learning of Object Keypoints for Perception and Control." Kulkarni et al., NeurIPS 2019.
> [6] "Interaction networks for learning about objects, relations and physics." Battaglia et al., 2016.
> [7] "Unsupervised Learning of Object Structure and Dynamics from Videos." Minderer, Matthias, et al. NeurIPS 2019.

---

### Official Review · Reviewer_YtC3 · 2021-11-03

**Correctness:** 3
**Technical Novelty And Significance:** 2
**Empirical Novelty And Significance:** 2
**Recommendation:** 3
**Confidence:** 4

**Main Review:**

**1. Strengths**

(1). The idea of combining unsupervised keypoint detection and forward modeling seems to be new and has not been done before.

(2). The idea is simple and shows its effectiveness to some previous baselines.

**2. Concerns**

The reviewer does have some concerns about the model.

(1). The claim on the previous method is not accurate enough that "Relying on object detection and segmentation tools, on the other hand, makes the forward model fragile and dependent on the flawless performance of these tools. More often than not, pretrained object detection or segmentation models suffer from poor generalization to unseen objects". Models based-on detection or segmentation proposals have not been compared in the experimental sections. Also, recent methods like MoNet[A] have been showing object proposals can be obtained from unsupervised learning. It will be interesting and necessary to compare the models based on keypoint detection and object proposals like combining the MoNet and PropNet for forward modeling.

(2). The evaluation is limited on a newly-built dataset and it will be more convincing to compare the proposed methods on the previous dataset like CLEVRER, which offers more baselines for future prediction.

(3). The details of the newly-built dataset is missing like the statistics of the dataset and what kind of objects it contains. It will be hard for readers to catch up how challenging the dataset is and how effective the model is.

(4). From the examples in Figure 3, the dataset seems to be simple. It contains simple objects from the top-down view.  whether the dataset contains occlusion cases? Is the camera stable? Much details about the dataset is needed.

[A]. Burgess C P, Matthey L, Watters N, et al. Monet: Unsupervised scene decomposition and representation[J]. arXiv preprint arXiv:1901.11390, 2019.


**Summary Of The Paper:**

**The main idea of the paper**

This paper combines two powerful ideas, 1. unsupervised keypoint extraction for objects and 2. object-centric representation for forward modeling. KINET first detects objects' key points in the video in an unsupervised manner and extract object-centric representation for future prediction. Experiments are conducted in a new dataset built on MuJoCo and the performance in table 1 shows the proposed framework achieves better performance than the baseline methods.

**Summary Of The Review:**

Since a bunch of significant baselines are absent and the details of the dataset are missing, the reviewer thinks the paper in the current version is not suitable for publishing on ICLR. The reviewer does think that a revised version will make the paper much more appealing.

---

> ### Author Response · Authors · 2021-11-25
> **Response to Reviewer YtC3**
>
> We thank the reviewer for their many constructive and insightful comments and suggestions. Several changes to the manuscript the been made including a new dataset and further experiments. We believe that through your assistance the paper has improved.
>
> ### Baseline Models
>
> > Models based-on detection or segmentation proposals have not been compared in the experimental sections.
>
>  We compare our approach with Interaction Network (IN) and Visual Interaction Network (VisIN) models. For these baselines, we are using the ground-truth object location and visual features from bounding boxes centered on the ground-truth object locations. We believe the performance of these two baseline models can be interpreted as an upperbound for other alternative baseline models with pretrained object-detection or segmentation model. On the other hand, even with an ideal object detection/segmentation model, the problem of hardcoding the number of objects in the formulation of object-centric graph-based forward models remains unsolved. Our approach avoids such hardcoding by learning a keypoint-based factorization of the system which leads to learning a more generalizable model.
>
>
> > ... recent methods like MoNet have been showing object proposals can be obtained from unsupervised learning. It will be interesting and necessary to compare with combining the MoNet and PropNet for forward modeling.
>
> Thank you for pointing out these papers. We think this is an interesting suggestion as MONet generalizes to an unseen number of objects and PorpNet is also a graph-based forward model. We would like to emphasize our interest in examining the proposed baseline model but we think the novel combination of MONet+PropNet would require further implementations and follow-up experiments which we are, unfortunately, unable to perform in this review process timeline.
>
> ### Dataset
>
>  > It will be more convincing to compare the proposed methods on the previous dataset like CLEVRER.
>
> This is an interesting suggestion. The CLEVRER dataset includes visual observations of interactions between multiple objects. Two main reasons that prevented us from using CLEVRER: (1) In this work, we focused on formulating an action-conditioned forward model. In the publicly available CLEVRER dataset, there is no information on the external action for the simulated episodes. (2) We were also interested in expanding the evaluation of our model's performance by incorporating our forward model in control tasks. The simulation environment that generated the CLEVRER dataset is not publicly released which holds back performing such control tasks.
>
>
> > The details of the newly-built dataset is missing.
>
> This is a valid concern. We apologize that we did not include this information. We added Appendix A to elaborate on the details of our MuJoCo simulation environment. This section gives an overview of the object geometries and dimensions in the training and generalization datasets.
>
> Our dataset will be released upon acceptance of the paper which can serve as a useful resource for future work with a focus on the action-conditional forward modeling. Following the recent free release of the latest version of MuJoCo, we are also planning to release the MuJoCo simulation environments we used to generate the dataset.
>
> > ... the dataset seems to be simple.
>
> Thanks for highlighting this. We agree that overhead camera observations were simplistic. We added a side-view camera to our MuJoCo environment and generated a new 3D dataset (see Figure 3). For both camera angles (top-down and angled-view) the camera location is fixed and stable. We repeated all experiments in the paper for both top-view and angled-view. The results section is updated accordingly.

---

### Official Review · Reviewer_QJbD · 2021-11-03

**Correctness:** 2
**Technical Novelty And Significance:** 3
**Empirical Novelty And Significance:** 2
**Recommendation:** 3
**Confidence:** 4

**Main Review:**

The paper tackles a crucial problem of learning the keypoints and their interactions from visual observations without supervision. The proposed method also has some novelty, such as learning probabilistic graphs and using contrastive loss to train the forward dynamics model. However, I have some concerns listed below.

- The paper misses some important related work, making its contribution less clear. For example, OP3[1] and V-CDN[2] predict object masks / keypoints and use graph nets to model dynamics, also without supervision. Moreover, both methods can generalize to unseen number of objects, contradicting the claim in the paper that "factorizing the scene into object instances limits the generalization of forward models to scenarios with a different number of objects". I think these two methods should serve as baselines for this paper.
- The dataset used in this paper seems a bit simplistic. Although the paper claims that "In our work, we consider experimenting with more complex three-dimensional objects where the objects are randomized and replicate real-world object manipulation", the actual dataset is "obtained using an overhead camera", making the observations essentially in 2D. In comparison, the dataset in [1] seems more challenging.
- The learned keypoints in Figure 3 are mostly around object centers. This behavior is quite different from [3], where the keypoints can capture object morphology. This can reduce the usefulness of the learned keypoints.
- The paper proposes to infer probabilistic graph representations. However, the pipeline seems broken. For example, in Equation 5, the node-level variables $Z$ are learned by reconstructing the adjacency matrix $A$, but $A$ is unknown and also needs to be inferred. How do you learn to infer $A$? And is this loss still a valid lower bound?
- Also the probabilistic approach did not show much improvement over the deterministic approach (see Table 1)
- The paper claims that the proposed model can generalize to unseen number of objects. However, if we compare Table 1 and 2, the error for generalization is an order of magnitude larger ($\times 10^{-2}$ in Table 2 vs $\times 10^{-3}$ in Table 1). I do not think this result supports the claim.

**MINOR COMMENTS**

- When you compute the prediction error for keypoints, how do you obtain the groundtruth keypoint position?
- In GraphMPC, how do you obtain the goal state graph configuration?

[1] Entity Abstraction in Visual Model-Based Reinforcement Learning. Veerapaneni et al., CoRL 2019.

[2] Causal Discovery in Physical Systems from Videos. Li et al., NeurIPS 2020.

[3] Unsupervised Learning of Object Keypoints for Perception and Control. Kulkarni et al., NeurIPS 2019.

**Summary Of The Paper:**

The paper proposes a keypoint-based forward dynamics model for generalization to unseen number of objects. The model employs unsupervised keypoint extraction methods from the literature, infers probabilistic graph representations over the keypoints, and makes forward predictions by message passing. The model is evaluated on multi-object manipulation tasks. It is shown that the model prediction error is close to that of a graph net with access to groundtruth object positions. The model is also shown to converge to the goal configuration faster than the baselines.

**Summary Of The Review:**

I recommend reject, because several claims are wrong or not well supported, and the contribution over related work is not sufficiently discussed or demonstrated.

---

> ### Author Response · Authors · 2021-11-25
> **Response to Reviewer QJbD [1/2]**
>
> We thank the reviewer for their many thoughtful and insightful comments and suggestions. Changes to the manuscript the been made accordingly. We believe that through your assistance the paper has improved.
>
> ### Choices for Baseline Models
>
> > OP3 and V-CDN methods should serve as baselines for this paper.
>
> Thank you for bringing these papers to our attention. We agree that OP3, and V-CDN frameworks are potentially generalizable to unseen number of objects and are certainly interesting baselines to compare with our method.
>
> For V-CDN, however, we believe the core objective is discovering the causal structure of the system which is why the perception module is extensively pre-trained only on a fixed environment. Our model, on the other hand, is trained end-to-end on randomized environments with random object shape and appearance. For this reason, we chose standard graph-based baseline models that have been used in such randomized environments (for instance [1,2]).
>
> We also think our work improves on OP3 problem formulation especially in the observation module. In OP3, the scene is factorized into the underlying entities using a generative model of segmentation masks which is formulated by inferring the depth information (see Appendix A in [3]). Our method, uses keypoint features to reconstruct the scene appearance and does not rely on inferring the depth information. Additionally, the action space in the OP3 object manipulation (pick and place) is different from our model (pushing).
>
>
> ### Dataset
>
> > ... the actual dataset is "obtained using an overhead camera", making the observations essentially in 2D.
>
> This is a great point and we agree that overhead camera observations were simplistic and essentially 2D. We added a side-view camera to our MuJoCo environment and generated a new 3D dataset. We repeated all experiments in the paper for both top-view and angled-view. The Results section is updated accordingly. We also think our new Angled View dataset is comparable to the dataset that the reviewer is suggesting (used in [3]) with even more complex object geometries (see Figure 3 in the updated manuscript).
>
>  ### Keypoint Assignment
>
> > The learned keypoints in Figure 3 are mostly around object centers.
>
> This is a correct observation. Our training dataset included cube-shaped objects (see Appendix Figure 4) for which the keypoint extraction module possibly assigned the keypoints around the center of the object because the feature maps extracted from the center sufficed reconstructing the appearance of the object for the overhead camera images. However, this is not the case for the new Angled View observations (see Figure 3). The extracted keypoints of the 3D objects in the Angled View dataset are spread out more for both training geometries and generalization geometries.
>
> ### Probablistic Graph Representation
>
> > How do you learn to infer $A$?
>
> We apologize if some of our notations were confusing in the Methods section. We rewrote the ambiguous equations for more clarification. As mentioned in equation (2) the probabilistic adjacency matrix is inferred by a decoder that takes as input the node embeddings and their latent variables. Following [4], this decoder is a simple MLP with sigmoid activation that outputs the probability over the edges in the graph.
>
> > The probabilistic approach did not show much improvement over the deterministic approach.
>
> We agree that based on Table 1 results the probabilistic message passing method that we introduced had marginal improvement compared to the deterministic message passing. We extended our ablation experiments and found that, in comparison with deterministic message passing, the probabilistic message passing significantly improves generalization to unseen number of objects and unseen geometries (see Table 2).
>
>  > ... I do not think this result (comparison of Table 1, and 2) supports the claim (generalization performance).
>
>  We apologize for this confusion. This is in fact a typo in Table 2 header. Both order of magnitudes are $\times 10^{-3}$. Still, we think a comparison between Table 1 and 2 does not directly support the claim that our method is generalizable to unseen number of objects. This is because Table 1 is reporting 1-step prediction accuracy while Table 2 is reporting the MPC control task result where the goal configuration varies a lot from the initial configuration. To make this comparison easier, in the updated Table 2, we are also reporting the MPC result for 3 objects (which is the number of objects in the training dataset).
>
>
>
> [1] "Object-centric forward modeling for model predictive control.", Ye, Yufei, et al., CoRL 2020.
>
> [2] "Learning long-term visual dynamics with region proposal interaction networks.", Qi, Haozhi, et al., ICLR 2021.
>
> [3] Entity Abstraction in Visual Model-Based Reinforcement Learning. Veerapaneni et al., CoRL 2019.
>
> [4] "Variational Graph Auto-Encoders." Thomas N Kipf, Max Welling, NeurIPS 2016.

---

> > ### Comment · Reviewer_QJbD · 2021-11-29
> > **Thank you for your response**
> >
> > Overall, I think what the paper does is quite similar to V-CDN, and making it end-to-end does not seem to be a significant contribution.
> >
> > Regarding the inference of adjacency matrix, in Variational Graph Auto-Encoders, the groundtruth adjacency matrix is provided, so the decoder can be trained to reconstruct the groundtruth adjacency matrix. However, in the proposed model, the groundtruth adjacency matrix is unknown, so I am wondering how the decoder can be learned and what A is used for computing log p(A|Z) in the inference loss.

---

> ### Author Response · Authors · 2021-11-25
> **Response to Reviewer QJbD (cont) [2/2]**
>
> ### Minor Comments
>
> > how do you obtain the groundtruth keypoint position?
>
> To compute the forward model loss we minimize the distance between the predicted location of the keypoints $\hat{x}^k_{t+1}$ from the forward model ($f_{fwd}$) and the extracted keypoint $f_{kp}(I_{t+1})$ (see equation 5).
>
>  > how do you obtain the goal state graph configuration?
>
> We use our learned model and extract the goal scene graph representation. Specifically, we extract the goal keypoints (nodes) and their corresponding node embeddings which together we use as the goal state in the GraphMPC algorithm.

---

### Decision · Program_Chairs · 2022-01-20

**Decision:**

Reject

**Comment:**

This submission received four high-quality reviews. After the discussion period, all reviewers agreed that this submission is not strong enough to be accepted. Concerns include the novelty of the proposed method wrt related work and the limited experiments. The AC agrees. The AC also finds it disappointing that the authors didn't address the concerns on novelty or even discuss the related papers suggested by the reviewers in the revision.

The recommendation is reject.